# Towards Efficient Federated Learning: Layer-Wise Pruning-Quantization Scheme and Coding Design

**DOI:** 10.3390/e25081205

**Published:** 2023-08-14

**Authors:** Zheqi Zhu, Yuchen Shi, Gangtao Xin, Chenghui Peng, Pingyi Fan, Khaled B. Letaief

**Affiliations:** 1Department of Electronic Engineering, Tsinghua University, Beijing 100084, China; 2Beijing National Research Center for Information Science and Technology (BNRist), Tsinghua University, Beijing 100084, China; 3Wireless Technology Laboratory, Huawei Technologies, Shanghai 200121, China; 4Department of Electronic and Computer Engineering, Hong Kong University of Science and Technology, Hong Kong

**Keywords:** federated learning, model pruning, parameter quantization, code design, layer-wise aggregation, communication-computation efficiency

## Abstract

As a promising distributed learning paradigm, federated learning (FL) faces the challenge of communication–computation bottlenecks in practical deployments. In this work, we mainly focus on the pruning, quantization, and coding of FL. By adopting a layer-wise operation, we propose an explicit and universal scheme: FedLP-Q (federated learning with layer-wise pruning-quantization). Pruning strategies for homogeneity/heterogeneity scenarios, the stochastic quantization rule, and the corresponding coding scheme were developed. Both theoretical and experimental evaluations suggest that FedLP-Q improves the system efficiency of communication and computation with controllable performance degradation. The key novelty of FedLP-Q is that it serves as a joint pruning-quantization FL framework with layer-wise processing and can easily be applied in practical FL systems.

## 1. Introduction

### 1.1. Background and Motivation

With the advent of the big data era, data privacy and system efficiency have attracted great attention, which also brings unprecedented challenges to the field of artificial intelligence (AI). As the first attempt to solve the problem of data silos, federated learning (FL) has now emerged as an important and promising branch of distributed machine learning [1]. Without sharing private data with other clients, FL requires only the interaction of local model parameters for cooperative learning, which significantly improves the security and efficiency of intelligent distributed systems.

The employment of AI in communication networks is already considered as a core feature of 6G systems [2]. In particular, FL has shown its strong potential in combination with deep neural networks [3,4]. As a matter of fact, FL not only naturally adapts to different structures of multi-user networks with distributed data [5], but also enables intelligent collaboration for multi-agent systems [6]. In addition, as a privacy-preserving mechanism that permits unified model training on distributed data from multiple clients, FL can be adopted in numerous fields, such as finance, healthcare, AIoT, smart cities, etc. [7].

The development of this distributed learning paradigm faces multiple challenges. In FL scenarios, clients have greater control over distributed data and local devices. Thus, there exists significant heterogeneity in terms of both data and devices. The former causes a non-iid data distribution, which hinders the convergence and the performance of the global model [8]. The heterogeneity of local devices and settings also leads to strict restrictions on the local computation and transmission, resulting in the straggler effects, where the systems have to wait for the slowest client [9]. In addition, in some mobile applications with numerous clients, the bandwidth resources are more limited and the communication costs for parameter sharing are much higher [10]. As a result, communication–computation efficiency becomes one of the most significant challenges in the practical deployment of FL systems [11]. An intuitive idea is to deploy models of different complexities on each client, which is called model heterogeneity. However, model heterogeneity violates the original assumption of FL, where the global model is supposed to be identical to all clients. Therefore, the designs of efficient heterogeneous FL systems are still in the early stage.

### 1.2. Related Work

In order to handle these issues related to system efficiency, two basic solutions have been considered: model pruning and parameter quantization. These were first introduced in centralized machine learning, aiming to accelerate the training process and compress the deep learning models [12]. These two techniques are broadly combined to slim the model and compress the data volume of representation. Inspired by the above two concepts, similar mechanisms have been transferred into the context of FL, which are easy to implement based on the existing FL frameworks. Specifically, pruning performed before local training or uploading, and quantization conducted before model aggregation.

The pruning strategies of FL have been broadly studied in the literature. Caldas et al. proposed the first federated pruning framework in [13], where the global model is compressed into sub-models for local clients. Based on this scheme, the pruning settings and the spectrum allocation for wireless FL were investigated in [14,15]. These methods use random width-wise pruning and have to consider different pruning rules for the inner layers with different structures. These schemes were also challenged in [16], where the authors doubted the rationality of randomly dropping the inner layer parameters. Furthermore, several new pruning mechanisms were also developed. For example, Horvath et al. pointed out that the order of the parameters matters in pruning aggregation and should be taken into consideration [17]. Depth-wise schemes based on blocks and layers were also discussed in [18,19], which showed that model compression in FL should affects the functionality of different parts in local models. The FedLP schemes in [19] specifically provided common pruning solutions for model heterogeneity cases, which extends the FL scenarios. These recent studies suggested that the pruning mechanisms should be designed specially due to the characteristic aggregation operation in FL.

As for quantization, it is commonly treated as an independent procedure from pruning. QSGD, a classic quantization scheme proposed in [20], has been integrated into FL systems [21,22]. Some extensions and variants for communication-efficient scenarios were also developed in [23,24]. The theoretical analysis was studied in [25], and some universal coding schemes were adopted in [26]. In addition, Prakash et al. investigated optimized FL by combining pruning and quantization [27]. However, specific coding schemes that fit the aforementioned pruning and quantization are essentially required. As combined technologies for model compression, pruning-quantization-coding can be jointly and compatibly designed to improve the efficiency for practical FL systems.

### 1.3. Contribution and Organization

Inspired by the above studies, we mainly consider a compressed FL scheme with joint layer-wise model pruning and parameter quantization in this work, which is an extension of [19]. The main contributions of this work can be summarized as follows:•We propose FedLP-Q (the codes in this work are available at https://github.com/Zhuzzq/FedLP/tree/main/FedLP-Q), a joint pruning-quantization FL framework employing the layer-wise approach. FedLP-Q provides a simple but efficient paradigm to mitigate the communication loads and computation complexity in FL systems;•We develop two distinct FedLP-Q schemes for homogeneous and heterogeneous cases. The corresponding pruning strategies, quantization rules, and the coding designs are presented in detail. The theoretical analysis also shows the strengths of the coding scheme for FedLP-Q;•We carry out several experiments to evaluate the performance of FedLP-Q. The outcomes suggest that this layer-wise pruning-quantization mechanism significantly improves the system efficiency with controllable performance loss.

The remainder of this article is organized as follows. In Section 2, we introduce the preliminaries and the basic concept of layer-wise pruning-quantization. In Section 3, we illustrate the specific design of the proposed approach and its theoretical analysis. The detailed experimental results and more discussions are presented in Section 4. Finally, in Section 5, we conclude and show several potential directions for this work.

## 2. Preliminaries and FedLP-Q Framework

In this section, we introduce some preliminaries and the framework of FedLP-Q. Before that, for explicity, we list some basic notations in Table 1.

### 2.1. Basic FL

In a typical FL system with *N* distributed clients, each client *k* possesses its own dataset $D_{k}$ to train its local model $\theta_{k}$. To preserve the privacy of personal data, FL conducts local training and periodic model aggregation. Clients cooperate to update the global model $\bar{\theta }$ via parameter sharing, which improves the communication efficiency. As a standard rule, FedAvg [1] selects *K* clients as a temporary active group $P_{t}$ and aggregates their local gradients to update the global model. At each global round *t*, participator clients train their local models for $E_{l}$ epochs and upload the local gradients to the server. The global model is updated by model aggregation as follows:(1)$${\bar{\theta }}_{t+1}\leftarrow {\bar{\theta }}_{t}-\sum_{k\in P_{t}}\frac{\omega_{k}}{\sum_{m\in P_{t}}\omega_{m}}\cdot \eta_{k}g_{k,t},$$
where $g_{k,t}$ is the accumulated gradients of client *k* after $E_{l}$ epoch local training. $\omega_{k}=\frac{|D_{k}|}{\sum |D_{m}|}$ is the aggregation weights and $\eta_{k}$ is the learning rate of each client.

### 2.2. An Overview of the FedLP-Q Framework

As discussed above, the aggregation mechanism of FL makes it possible to conduct layer-wise processes for local models. Furthermore, FL with layer-wise pruning has shown its strengths in several aspects including stability, convergence, and communication–computation efficiency [19]. We generalize such schemes by adding the quantization for layer-wise pruning and formulate the FL framework with layer-wise pruning-quantization, abbreviated as FedLP-Q.

Consider a full model with *L* layers, $\theta :=\left[\theta^{1},\theta^{2},\cdots ,\theta^{L}\right]$. After a round of local training on client *k*, its accumulated gradients can be represented by:(2)$${\tilde{g}}_{k}=\left[g_{k}^{l}\right],\;\;l\in L_{k},$$
where $L_{k}$ is the index set of the preserved layers and is determined by the specific layer-wise pruning strategies. The pruned accumulated gradients of each layer are further quantized into $Q(g_{k}^{l})$, where Q(·) denotes the layer-wise quantization function. Hence, the layer-wise aggregation rule shall be modified as:(3)$${\bar{\theta }}_{t+1}^{l}\leftarrow {\bar{\theta }}_{t}^{l}-\sum_{k\in P_{t}}\frac{1_{k}^{l}\cdot \omega_{k}\eta_{k}}{\sum_{m\in P_{t}}1_{m}^{l}\cdot \omega_{m}}Q\left(g_{k,t}^{l}\right),$$
where $1_{k}^{l}$ is the probabilistic indicator function with value {0,1} to represent whether the *l*-th layer is preserved for the local model on client *k*. The examples of the indicators will be presented in the following section.

Figure 1 shows the basic frameworks of FedLP-Q for homogeneous and heterogeneous local model scenarios. In the homogeneity scheme, all clients possess the same full layers of the global model. In the heterogeneity scheme, clients initialize their local pruned models with different sequential layers. During the aggregation rounds, clients prune their local models according to layer-wise pruning. Then, the sub-models are quantized into lower bit rates and encoded for model uploading. The parameter server decodes and aggregates the quantized gradients. Similarly, the updated global model will be encoded and transmitted to each local client. In FedLP-Q, since each layer of the model is the basic unit, the pruning and quantization processes can be explicit and model-agnostic. For the above reasons, layer-wise pruning-quantization schemes require no previous knowledge of the model and can easily be deployed in FL systems.

## 3. FedLP-Q: Quantization-Coding Scheme for Layer-Wise Pruning

In this section, we will introduce the proposed layer-wise pruning-quantization approach and the coding scheme in detail. The convergence and the coding performance of FedLP-Q will also be analyzed.

### 3.1. Pruning Phase

To develop layer-wise pruning for FL, we shall consider the homogeneous and heterogeneous local models. Similar to [19], we adopt the probabilistic layer-wise pruning strategy for homogeneity cases. For heterogeneity cases, we assign the pruned sub-models with different sequential layers to the clients.

#### 3.1.1. Homogeneity Case

As shown in Figure 1a, all clients possess the same full model. While finishing the local training, clients carry out layer-wise pruning for the accumulated gradients. Probabilistic layer-wise pruning proceeds as follows: before uploading the accumulated gradients for aggregation, each client *k* preserves the gradients of the *l*-th layer with probability $p_{k}^{l}$, termed as the layer-preserving rate (LPR). In other words, the indicators in (Equation 3) satisfies:(4)$$1_{k}^{l}=\left\{\begin{array}{ccc} 1 & \;\;\mathrm{with}\;\mathrm{probability} & \;p_{k}^{l},\\ 0 & \;\;\mathrm{with}\;\mathrm{probability} & \;1-p_{k}^{l}. \end{array}\right.$$
The pruned accumulated gradients ${\tilde{g}}_{k}$ in (Equation 2) for homogeneity cases can be formulated according to $\left\{1_{k}^{l}\right\}$.

#### 3.1.2. Heterogeneity Case

The major challenge of FL with heterogeneous local models is to design the rules for model assignment and aggregation. Fortunately, the layer-wise mechanism of FedLP-Q gives an explicit solution to such problems. That is, each client initializes the local model of different layers. Let $L_{k}$ denote the layer count (LC). As shown in Figure 1b, its local model shall be initially pruned into ${\tilde{\theta }}_{k}=\left[\theta_{k}^{1},\cdots ,\theta_{k}^{L_{k}},\left(\theta_{k}^{O}\right)\right]$, where $\theta_{k}^{O}$ (if it exists) is the personalized fully connected (FC) layer to fit the output dimensions and will not be uploaded for aggregation. Then, the pruned accumulated gradients of client *k* in heterogeneity settings can be expressed as:(5)$${\tilde{g}}_{k}=\left[g_{k}^{1},\cdots ,g_{k}^{L_{k}}\right].$$
and the layer indicators $\left\{1_{k}^{l}\right\}$ can be determined.

In this case, the computation complexity for local training is notably reduced since client *k* only needs to train a sub-model with $L_{k}$ layers. As a result, the straggler clients are able to contribute to the global model by training and transmitting small sub-models. This scheme considers the imbalance of the local devices and their computation–communication capabilities, which is more practical in real scenarios.

### 3.2. Quantization and Coding Phase

Based on layer-wise pruning, we further formulate the corresponding quantization settings and the coding scheme.

#### 3.2.1. Quantization Scheme

For each preserved layer, FedLP-Q conducts the quantization method with equal intervals, which is similar to the quantized-SGD (QSGD) in [20]. Layer-wise quantization after pruning proceeds as follows: for a preserved layer *l* on client k and the preset quantization bit *b*, the quantized aggregated gradient $Q(g_{k}^{l};b)$ is defined by:(6)$$Q\left(g_{k}^{l};b\right)=\frac{{∥g_{k}^{l}∥}_{2}}{2^{b}}\cdot \mathrm{sgn}\left(g_{k}^{l}\right)⊙\mathrm{seg}\left(g_{k}^{l};b\right),$$
where ⊙ is the element-wise product, sgn(·) is the sign function, and $\mathrm{seg}\left(g_{k}^{l};b\right)$ is the interval segmentation function. Specifically, for the full gradients $g_{k}^{l}$ of the *l*-th layer, we firstly divide its $\ell_{2}$-norm into $2^{b}$ equal intervals, which takes *b* bits. Then, each gradient in the layer follows a stochastic quantization: for $g_{k}^{\left(l,i\right)}$, if it locates at the *s*-th interval, i.e., $|g_{k}^{\left(l,i\right)}|/∥g_{k}^{l}{∥}_{2}\in \left[\frac{s}{2^{b}},\frac{s+1}{2^{b}}\right]$, the gradient $g_{k}^{\left(l,i\right)}$ will be quantized as the values of two sides according to the distances from the two endpoints of the interval. The quantization can be expressed as:(7)$$\mathrm{seg}\left(g_{k}^{\left(l,i\right)};b\right)=\left\{\begin{array}{cc} s+1 & \mathrm{with}\;\mathrm{probability}\;p_{Q}^{\left(l,i\right)}\left(s;b\right),\\ s & \mathrm{otherwise}. \end{array}\right.$$
Therein, the stochastic quantization probability is defined as:(8)$$p_{Q}^{\left(l,i\right)}\left(s;b\right)=\frac{|g_{k}^{\left(l,i\right)}|}{∥g_{k}^{l}{∥}_{2}}\cdot 2^{b}-s,$$
which is the ratio between the distance to the left endpoint and the interval length.

#### 3.2.2. Coding Design

Given the quantization bit *b* and the layer 2-norm $∥g_{k}^{l}{∥}_{2}$, the quantized value of each gradient can be handled as the index of the interval. Then, the coding of the quantized gradients can be considered as the representation of the integers. Therefore, we adopt a classic solution: recursive Elias coding, which is also known as Elias omega coding  [28]. The core idea of Elias omega coding is to recursively encode the prefix of the binary representation. Hence, the code length of any positive integer *N* follows |Elias(N)|=1+logN+loglogN+⋯=1+(1+o(1))logN. Then, we shall consider the simple encoding scheme for layer-wise pruning-quantization. The full coding scheme for FedLP-Q proceeds as follows: For each preserved layer *l*, the layer 2-norm will firstly be encoded into 32 bits (float). The quantization setting *b* (integer) will be attached using Elias omega coding. Then, for each gradient, the index of interval ${\mathrm{seg}}_{k}^{l}\left(i\right)$ for stochastic quantization will be sequentially encoded according to the Elias omega coding. The sign of each gradient will additionally take 1 bit (a boolean variable standing for positive/negative). The decoding scheme is also simple: The decoder firstly reads off the 32 bits to obtain the layer 2-norm $∥g_{k}^{l}{∥}_{2}$ and decodes the Elias codes of *b*. Then, for each gradient, its interval index will be reconstructed according to Elias omega decoding, while the sign will be decoded by the last bit.

After layer-wise pruning-quantization, the layer gradients can be represented by a 4-element tuple $\left\{∥g_{k}^{l}{∥}_{2},\;b,\;\mathrm{seg}\left(g_{k}^{l};b\right),\;\mathrm{sgn}\left(g_{k}^{l}\right)\right\}$. Therefore, based on Elias omega coding, the code structure of each layer with $M_{l}$ parameters is shown in Figure 2. Furthermore, the corresponding encoding and decoding procedures can be developed as shown in Algorithm 1 and Algorithm 2, respectively.
**Algorithm 1** Encoder for layer *l*
**Input:**$\left\{∥g_{k}^{l}{∥}_{2},\;b,\;\mathrm{seg}\left(g_{k}^{l};b\right),\;\mathrm{sgn}\left(g_{k}^{l}\right)\right\}$, $M_{l}$;

   1: $C_{l}\leftarrow \mathrm{Concat}\left(\mathrm{Float}32\left(∥g_{k}^{l}{∥}_{2}\right),\mathrm{Elias}\_\mathrm{Encoder}\left(b\right)\right)$;
▹encode the header   2: **for**
 $i\leftarrow 1\;\mathrm{to}\;M_{l}$ 
**do**

   3:    $C_{l}\leftarrow \mathrm{Concat}\left(C_{l},\mathrm{Elias}\_\mathrm{Encoder}\left(\mathrm{seg}(g_{k}^{\left(l,i\right)};b)\right)\right)$;
▹ encode the interval index   4:    $C_{l}\leftarrow \mathrm{Concat}\left(C_{l},\mathrm{Binary}\left(\mathrm{sgn}(g_{k}^{\left(l,i\right)};b)\right)\right)$;▹ encode the gradient sign   5: **end for**
**Output:** encoded message $C_{l}$.


**Algorithm 2** Decoder for layer *l*
**Input:** received message $C_{l}$; **Initialization:** ${\mathrm{seg}}^{l}\leftarrow \left[\;\right]$, ${\mathrm{sgn}}^{l}\leftarrow \left[\;\right]$, i←0;
   1: $G_{l}\leftarrow \mathrm{Float}32\_\mathrm{Decoder}\left(C_{l}\right)$; ▹decode the gradient norm   2: 
$C_{l}>>32$;
   3: $\left(b,l_{E}\right)\leftarrow \mathrm{Elias}\_\mathrm{Decoder}\left(C_{l}\right)$;▹decode the quantization bit   4: $C_{l}>>l_{E}$;
   5: **while**
 $C_{l}$ 
**do**
   6:     $\left({\mathrm{seg}}_{i}^{l},l_{E}\right)\leftarrow \mathrm{Elias}\_\mathrm{Decoder}\left(C_{l}\right)$;▹ decode the interval index   7:     $C_{l}>>l_{E}$;
   8:     $\left({\mathrm{sgn}}_{i}^{l},l_{E}\right)\leftarrow C_{l}$;▹ decode the gradient sign   9:     i←i+1; $C_{l}>>1$;
  10: **end while**
**Output:** decoded elements $\left\{G_{l},\;b,\;{\mathrm{seg}}^{l},\;{\mathrm{sgn}}^{l}\right\}$.


#### 3.2.3. Algorithm Formulation

Finally, we formulate the procedures of FedLP-Q as in Algorithm 3. The main changes from the original FL include (1) layer-wise pruning (for both the homogeneity and heterogeneity case ) for the local gradients is conducted in Line 5; (2) stochastic quantization and the corresponding encoding/decoding are performed in Lines 6 and 7; and (3) Line 9 introduces the layer-wise aggregation for the proposed pruning-quantization schemes.
**Algorithm 3** FedLP-Q
**Initialization:** local models, pruning configures, etc.
   1: **for**
t←1tomax_epoch
**do**

   2:     **for** participator client *k* in parallel **do**
▹ client side   3:         Update $\theta_{k,t}\leftarrow \mathrm{Local}\_\mathrm{Train}\left(\theta_{k,t-1};D_{k}\right)$; 
   4:         Calculate the accumulated gradients $g_{k,t}$; 
   5:         Conduct layer-wise pruning: ${\tilde{g}}_{k,t}$; 
   6:         Quantize the local gradients: $Q({\tilde{g}}_{k,t})$; 
   7:         Carry out Algorithm 1 to encode the quantized gradients; 
   8:         Upload local message to parameter server; 
   9:     **end for**

  10:     Carry out Algorithm 2 to decode the received messages; ▹ server side  11:     Aggregate each layer ${\bar{\theta }}_{t}^{l}$ by (Equation 3); 
  12:     Download the global model: $\theta_{k,t}\leftarrow {\bar{\theta }}_{t}$;▹ client side
  13: **end for**

**Output:** global model: ${\bar{\theta }}_{t}$.


### 3.3. Theoretical Analysis

Then, we theoretically analyzed the convergence and coding performance of FedLP-Q.

#### 3.3.1. Impacts on Convergence

Since FedLP-Q modifies the aggregation process of the original FL, the main impacts on convergence come from layer gradient pruning and quantization. Assuming that local training is independent from the pruning operations, the following proposition indicates the convergence result of FedLP-Q.

**Proposition** **1.**
*Consider a fairness case where $\omega_{k}=1/K$ and $p_{k}^{l}=p$ for all k=1,⋯,K. FedLP-Q leads to ${\left(1-p\right)}^{K}$ convergence rate decay compared to the original FL scheme, i.e., the expectation of the aggregated gradient in FedLP-Q satisfies:*

(9)
$$E\left[{\hat{g}}^{l}|{\bar{g}}^{l}\right]=\left[1-{\left(1-p\right)}^{K}\right]\cdot {\bar{g}}^{l}.$$

*Therein,*

(10)
$${\bar{g}}^{l}=\frac{1}{K}\sum_{k}g_{k}^{l},\;{\hat{g}}^{l}=\sum_{k}\frac{1_{k}^{l}}{\sum_{m}1_{m}^{l}}Q\left(g_{k}^{l}\right)$$

*denote the aggregated gradients in the original FL and FedLP-Q, respectively.*


**Proof.** For simplicity, rewrite the client index in $P_{t}$ as {1,2,⋯,K}. According to the layer-wise aggregation in (Equation 3), the expectation of the pruned-quantized gradients can be obtained by:
(11)$$\begin{array}{cc} E\left[{\hat{g}}^{l}|{\bar{g}}^{l}\right] & =E\left[\sum_{k=1}^{K}\frac{1_{k}^{l}}{\sum_{m}1_{m}^{l}}Q\left(g_{k}^{l}\right)|{\bar{g}}^{l}\right]=\sum_{k=1}^{K}E\left[\frac{1_{k}^{l}}{\sum_{m}1_{m}^{l}}Q\left(g_{k}^{l}\right)|{\bar{g}}^{l}\right] \end{array}$$
(12)$$\begin{array}{cc} \; & \overset{➀}{=}\sum_{k=1}^{K}E\left\{E\left[\frac{1_{k}^{l}}{\sum_{m}1_{m}^{l}}Q\left(g_{k}^{l}\right)|{\bar{g}}^{l},1_{k}^{l}\right]|{\bar{g}}^{l}\right\} \end{array}$$
(13)$$\begin{array}{cc} \; & =\sum_{k=1}^{K}p\cdot E\left[\frac{1}{1+\sum_{m\neq k}1_{m}^{l}}\cdot Q\left(g_{k}^{l}\right)|{\bar{g}}^{l}\right] \end{array}$$
(14)=∑k=1Kp∑m=1K1mK−1m−1pm−1(1−p)K−mEQ(gkl)|g¯l
(15)=➁∑k=1K∑m=1K1KKmpm(1−p)K−mEQ(gkl)|g¯l
(16)$$\begin{array}{cc} \; & =\left[1-{\left(1-p\right)}^{K}\right]E\left[\sum_{k=1}^{K}\frac{Q(g_{k}^{l})}{K}|{\bar{g}}^{l}\right] \end{array}$$
(17)$$\begin{array}{cc} \; & \overset{➂}{=}\left[1-{\left(1-p\right)}^{K}\right]\cdot {\bar{g}}^{l}, \end{array}$$
where ➀ is the result of equation $E\left[X|Y\right]=E\left\{E[X|Y,Z]|Y\right\}$, ➁ holds because 1mK−1m−1=(K−1)!(K−m)!m!=1KKm, and ➂ is the unbiased quantization property of QSGD-based quantizers [22,25]. Then, the proposition is proved. □

**Remark** **1.**
*(Equation 9) shows that the aggregated gradients of FedLP-Q decreases compared to the original FL schemes, which implies ${\left(1-p\right)}^{K}$ convergence rate decay. In addition, the conclusion of Proposition 1 can be directly adopted in the assumptions of model pruning/quantization (e.g., [22,24]) and the specific convergence theorems can be further modified.*


According to the proposition, such impacts on convergence can be mitigated by increasing either the layer preserving rate *p* or the number of participator clients *K*.

#### 3.3.2. Coding Performance

Based on the stochastic quantization method and the corresponding coding scheme, we obtained the following proposition, which highlights the communication costs of FedLP-Q.

**Proposition** **2**(Code length)**.**
*By employing layer-wise pruning-quantization, the average bit number to communicate $g_{k}^{l}$ for client k is upper-bounded by:*
(18)$$\hat{C}\left(g_{k}^{l};b\right)=33+\left(1+o(1)\right)\log b+M_{l}\left[2+\left(1+o(1)\right)b\right].$$

**Proof.** The proposition is an intuitive result of the proposed quantization coding scheme. As discussed above, the code length of an integer *N* using Elias omega coding satisfies |Elias(N)|=1+logN+loglogN+⋯=1+(1+o(1))logN. Consider the code blocks of the *l*-th layer as shown in Figure 2, the code length can be divided into three parts: (1) The first block to represent the layer 2-norm $∥g_{k}^{l}{∥}_{2}$ takes 32 bits; (2) the Elias codes of the quantization level *b* takes 1+(1+o(1))logb bits; and (3) after stochastic quantization, the interval index of each parameter is an integer less than $2^{b}$, which takes 1+(1+o(1))b bits, and its sign takes one more bit. Hence, the total code length can be upper bounded as shown in (Equation 18). □

**Remark** **2.**
*According to [20], the original coding scheme proposed in QSGD for the l-th layer leads to the average code length asymptotically bounded by:*

(19)
$$32+\left[3+\left(1.5+o(1)\right)\log\frac{2(S^{2}+M_{l})}{S(S+\sqrt{M_{l}})}\right]S\left(S+\sqrt{M_{l}}\right),$$

*where $S=2^{b}$ is the count of the quantization intervals. Comparing the two upper bounds of encoding the layer-wise gradients in (Equation 18) and (Equation 19), one can find that the proposed coding scheme for FedLP-Q has a shorter code length than that in the original QSGD when the number of parameters is not extremely large.*


Specifically, the Elias coding approach for QSGD considers the positional coding of the non-zero gradient to fit the cases with sparse and numerous parameters. On the contrary, in FedLP-Q, the gradients of each layer are handled independently. Therefore, the sparsity and the element amounts are not enough to show the advantages of the Elias coding scheme in QSGD. The experimental results in the next section also indicate that the quantization coding for FedLP-Q can be simple but more efficient. Overall, by employing layer-wise pruning and quantization, FedLP-Q improves the communication–computation efficiency in FL, and also simplifies the code design for model uploading.

Furthermore, we discuss the code efficiency. After layer-wise pruning-quantization, the gradients of $Q(g_{k}^{l})$ can be represented by a set of interval indexes. Then, the entropy rate of the layer gradients can be defined as follows:

**Definition** **1.**
*After the b-bit quantization, the entropy rate of the layer gradients $g_{k}^{l}$ is defined as:*

(20)
$$H\left(Q(g_{k}^{l});b\right):=\lim_{n\to \infty }\frac{1}{n}\sum_{i=1}^{n}H\left({\mathrm{seg}}_{k}^{l}\left(i\right)\right);$$

*where seg(·) is the quantization index. Consider the independence of each gradient and the empirical distribution, the above entropy rate can be approximated by:*

(21)
$$\hat{H}\left(Q(g_{k}^{l});b\right)\approx \hat{H}\left({\mathrm{seg}}_{k}^{l}\right):=-\sum_{s\in \{{\mathrm{seg}}_{k}^{l}\}}f\left(s\right)\log\left(f(s)\right),$$

*where $f\left(s\right)=\frac{\mathrm{count}(s)}{M_{l}}$ is the proportion of index s in ${\mathrm{seg}}_{k}^{l}$.*


Then, the code efficiency can be investigated by comparing the per-gradient code length and the entropy rate. According to Proposition 2, the per-gradient code length can be expressed asymptotically by:(22)$$\frac{\hat{C}\left(g_{k}^{l};b\right)}{M_{l}}\approx 2+\left(1+o(1)\right)b.$$
Hence, the aim is to make the asymptotic per-gradient code length as close as possible to the entropy rate, i.e., $\frac{\hat{C}\left(g_{k}^{l};b\right)}{M_{l}}\to \hat{H}\left(Q(g_{k}^{l});b\right)$. Moreover, consider the whole procedure of FedLP-Q, the equivalent per-gradient code length $p_{k}^{l}\frac{\hat{C}\left(g_{k}^{l};b\right)}{M_{l}}$ can be even smaller than the entropy rate $\hat{H}\left(Q(g_{k}^{l});b\right)$, where $p_{k}^{l}$ is the preserving proportion of the gradients of the *l*-th layer. In the next section, we will show that the proposed FedLP-Q can achieve the above goals through experiments.

## 4. Experimental Results

In this section, we describe several experiments to evaluate the performance of the FedLP-Q schemes.

We carried out the experiments on an image classification FL task on CIFAR-10 with 100 clients and a 0.1 participation rate. Specifically, the parameter server randomly selected 10 clients for aggregation in every global round. The number of the local training epoch was fixed at 5. We adopted the global model as a CNN-based neural network with six Conv layers and two FC layers. Batch normalization and maxpooling were also conducted following each Conv layer. As for the two proposed schemes, we adopted the consistent LPR *p* for the homogeneous case, abbreviated as FedLP-Q_Homo(*p*). For the heterogeneous case, we separated the Conv layers into five ordered sequences with LC from 1 to 5. These five sub-models were assigned to clients according to LC distributions. FedLP-Q_Hetero(*l*) means that the model with *l* LC was assigned with the highest probability and the parameter ’u’ represents the uniform assignment.

We evaluated the performance of the proposed FedLP-Q schemes under both iid and Dirichlet non-iid data (α = 1) settings. As shown in Figure 3, the FedLP-Q schemes reached a similar accuracy and convergence as the original FedAvg. In particular, the homogeneous case with high quantization bits (*b* = 10) led to nearly the same test accuracy curves as that of FedAvg, but significantly reduced the communication traffic. Small quantization bits and the non-iid data settings led to a lower accuracy and the instability of the global model. Therefore, there exists redundancy in the transmitted gradients and the quantization representation. By setting different FedLP-Q configures, FL systems can reduce the requirement of the communication rate and computation capability, which enables more devices to participate in the FL tasks.

The detailed numerical results for the test accuracy and the communication–computation efficiency are listed in Table 2. The savings of communication and computation are represented by the decay ratios of the parameter count and the million floating point of operations (MFLOPs) per local model, respectively. One can find that, for homogeneity cases, FedLP-Q performs better on both data settings. The randomness caused by the probabilistic layer-wise pruning and stochastic quantization even benefit the model performance. In addition, the number of parameters can be reduced, which relieves the communication loads. For heterogeneity cases, though there exists degradation of the test accuracy, FedLP-Q can significantly reduce both the communication and computation costs with acceptable model performance. This is an important characteristic, especially for mobile scenarios where the clients’ capabilities of transmission and local computation are limited.

Figure 4 displays the model performance, communication traffic, and computation load savings of several schemes, including FedAvg, homogeneous FedLP-Q, and heterogeneous FedLP-Q, with different quantization settings. The higher vertical axis represents a higher model accuracy. The x-axis denotes the reduced communication traffic (MB) and the y-axis is the saved local computation complexity (MFLOPs). The horizontal plane implies the communication and computation efficiency. If a projection on the x-y plane lies farther from the zero point, the scheme saves more communication and computation loads. By plotting the 3D figure, the trade-offs between the model performance and system efficiency can be intuitively shown. According to this trade-off, the corresponding system design and the pruning-quantization settings can be well guided.

We next evaluated the efficiency of the coding scheme for FedLP-Q. Figure 5a shows how the code length evolves as the number of the parameters increases. We compared the expected code length of the proposed Elias coding for FedLP-Q (as in (Equation 18)), the Elias coding for QSGD (as in (Equation 19)), and the non-quantization coding (Float32). For a parameter count of less than 600 K, the Elias-based coding scheme proposed in Section 3.2.2 led to the lowest expected code length. The superiority of the QSGD coding scheme in [20] appeared for an extremely large parameter count. Notably, since we adopted a layer-wise approach, the parameter count of each layer commonly fell within the red region marked in Figure 5b. As a result, the proposed coding scheme is more suitable for FedLP-Q. Figure 5b presents the practical code length of the neural network model employed in the experiments. By implementing the Elias code, FedLP-Q encodes the accumulated gradients into a lower bit rate, which reduces the requirement for the communication rate in real-world systems.

Finally, we investigated the gap between the per-gradient length of the proposed schemes and the entropy rate of the gradients. Figure 6 displays the quantized gradient distribution of the selected clients under iid and non-iid data settings. The quantized gradient distributions of different layers have similar shapes under both iid and non-iid settings. This again demonstrates the potential and future directions of introducing layer-wise operation into FL. That is, the distribution and the entropy of each layer gradients can be modeled. Then, the communication traffic of each client can be better estimated and further coding schemes can be specifically designed.

Figure 7 shows the entropy rate $\hat{H}\left(Q(g_{k}^{l});b\right)$ and the equivalent per-gradient code length $p_{k}^{l}\frac{\hat{C}\left(g_{k}^{l};b\right)}{M_{l}}$ of each layer. Herein, the entropy rate of the gradients can be calculated through (Equation 21) based on the distributions shown in Figure 6. It is intuitive that the per-gradient length of the proposed schemes is close to the entropy rate of the gradients. In particular, as discussed in Section 3.3.2, with a lower LPR, the expectation of the per-gradient code length can take even fewer bits than the entropy rate, e.g., as shown by the curves of $p_{k}^{l}=0.7$. This indicates that the proposed coding scheme is efficient and can be adjustable to meet different system requirements on compression.

All the above results suggest that there exists a significant redundancy in the interaction of FL systems. Layer-wise pruning-quantization and the proposed coding design can be applied without damaging the model performance. Therefore, it can be concluded that FedLP-Q is an efficient scheme that improves the system efficiency and facilitates the deployment of FL applications by mitigating the straggler effects.

## 5. Conclusions

In this article, we proposed FedLP-Q, an efficient pruning-quantization approach for FL by adopting layer-wise operations. The corresponding pruning strategies, stochastic quantization, and coding scheme were formulated. Both the theoretical and experimental analyses indicated that FedLP-Q achieves a better communication–computation efficiency with controllable performance loss. This may be vital for the deployment of FL applications in mobile scenarios.

This work also opens up several potential research directions for future work, including exploring the robustness and security of layer-wise pruning-quantization, combinations with other FL structures, quantization/coding designs based on layer-gradient entropy estimation, and more practical modeling for the communication of FL clients.

## Figures and Tables

**Figure 1 entropy-25-01205-f001:**
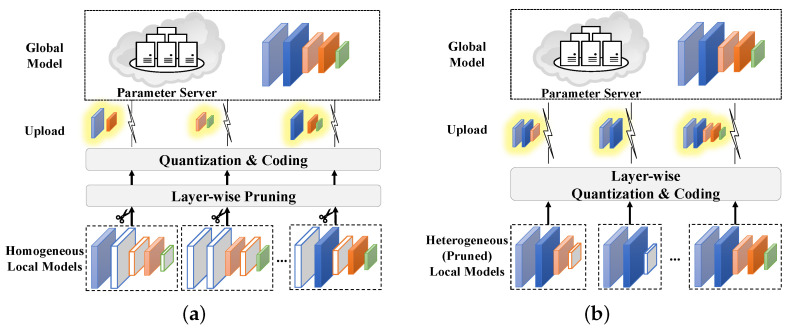
Two typical FedLP-Q schemes for different local model settings. The inactive layers (in gray) are removed from the sub-models for uploading. (**a**) Homogeneity scheme. (**b**) Heterogeneity scheme.

**Figure 2 entropy-25-01205-f002:**
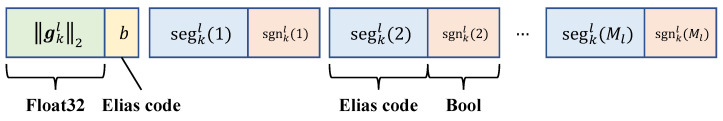
Code structure of each layer’s gradients under the FedLP-Q scheme.

**Figure 3 entropy-25-01205-f003:**
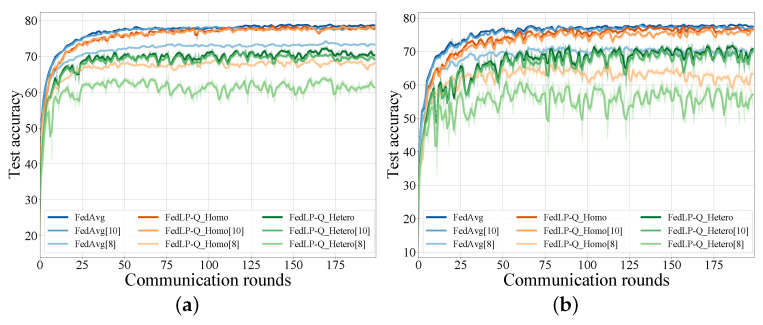
Test on CIFAR-10. FedLP-Q_Homo(0.5) and FedLP-Q_Hetero(u). (**a**) Under iid data. (**b**) Under non-iid data.

**Figure 4 entropy-25-01205-f004:**
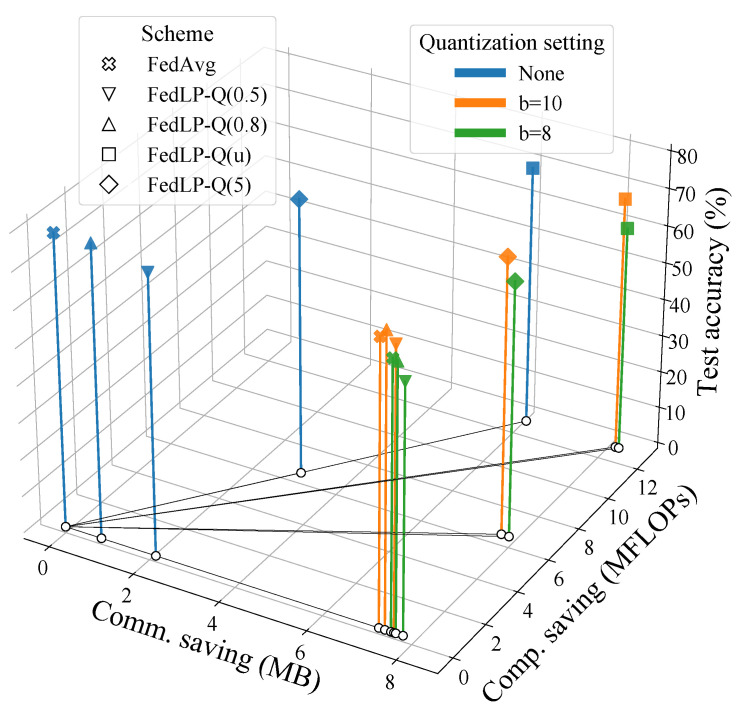
Trade-off: accuracy vs. communication–computation savings (under iid data).

**Figure 5 entropy-25-01205-f005:**
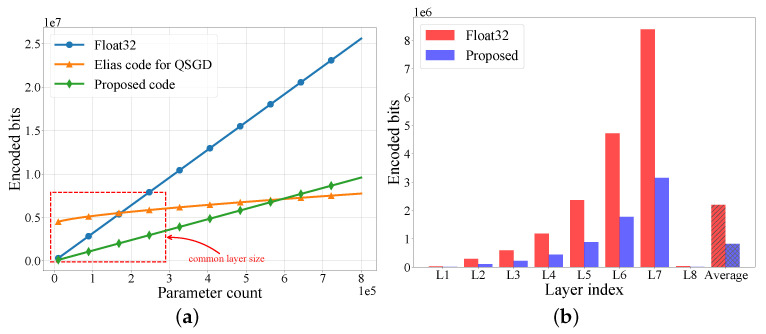
The comparison of coding schemes of model uploading (*b* = 10). (**a**) Encoded bits vs. parameters. (**b**) Code length of each layer.

**Figure 6 entropy-25-01205-f006:**
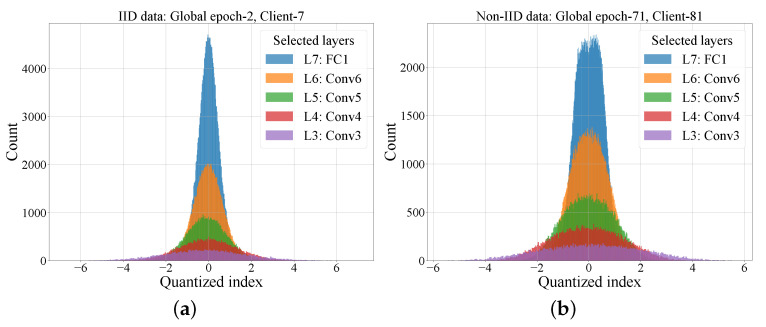
The quantized gradient distribution of selected clients. (**a**) Under iid data, 2nd global epoch, client 7. (**b**) Under non-iid data, 71st global epoch, client 81.

**Figure 7 entropy-25-01205-f007:**
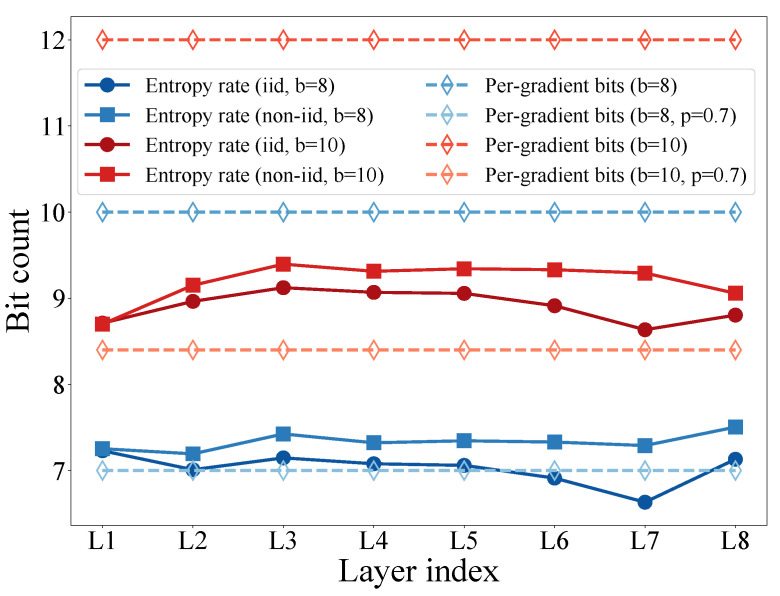
The entropy rate $\hat{H}\left(Q(g_{k}^{l});b\right)$ and the equivalent per-gradient code length $p_{k}^{l}\frac{\hat{C}\left(g_{k}^{l};b\right)}{M_{l}}$ of each layer.

**Table 1 entropy-25-01205-t001:** Basic Notations.

Notations	Description
*N*	The number of distributed clients.
$E_{l}$	Local training epoch.
$\left\{D_{1},\cdots ,D_{N}\right\}$	Distributed local datasets.
$P_{t}$	The set of participation clients at *t*-th global epoch.
*K*	The number of participation clients.
$\bar{\theta }$	The global model.
$\left\{\theta_{1},\cdots ,\theta_{N}\right\}$	Local models.
$\left\{\omega_{1},\cdots ,\omega_{N}\right\}$	Aggregation weights for each client.
$g_{k,t}$	The accumulated gradient of client *k* in *t*-th global epoch.

**Table 2 entropy-25-01205-t002:** Comparisons on accuracy, communication, and computation saving.

Schemes	Test Accuracy % (iid/Non-iid)			#Param ↘	MFLOPs ↘
	No Quant	b = 10	b = 8		
FedAvg	78.8/75.3	76.2/74.9	**71.7**/**68.5**	/	/
FedLP-Q(0.2)	73.4/68.2	71.5/66.8	56.6/50.4	40.0%	/
FedLP-Q(0.5)	75.5/73.5	75.4/72.5	66.7/62.0	25.0%	/
FedLP-Q(0.8)	**78.9**/**77.3**	**78.3**/**76.2**	71.3/66.9	10.0%	/
FedLP-Q(1)	64.4/60.3	64.1/59.2	53.6/49.4	**84.6%**	**51.2%**
FedLP-Q(3)	66.8/62.0	66.1/61.7	58.4/53.9	79.6%	32.3%
FedLP-Q(u)	70.1/66.5	68.3/65.4	60.7/55.7	71.1%	34.5%
FedLP-Q(5)	74.6/73.1	74.4/72.1	68.6/64.5	35.6%	17.2%

## Data Availability

Not applicable.

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
