# Peer review of "Towards Efficient Federated Learning: Layer-Wise Pruning-Quantization Scheme and Coding Design"

_entropy, 2023, doi:10.3390/e25081205_

Round 1

Reviewer 1 Report

Summary: Two basic solutions, model pruning and parameter quantization, have been adapted from centralized machine learning to federated learning (FL), aiming to speed up the training process and compress the models. This paper introduces a scheme, FedLP-Q, that applies layer-wise model pruning and parameter quantization jointly to enhance the efficiency of FL systems. It details two different FedLP-Q schemes for homogeneous and heterogeneous scenarios, specifying the corresponding pruning strategies, quantization rules, and coding designs. The paper concludes that the layer-wise pruning-quantization mechanism significantly improves system efficiency while keeping performance losses acceptable.

This paper is well-structured and well-written, providing theoretical contributions. The numerical results validate the proposed framework. Here is my comment to improve the paper.

Major Comment: There exist many comparison techniques. Although this paper compared with federated averaging, there are also techniques that prune width-wise (e.g., slimmable neural networks). What is the reason for using a depth-wise pruning technique? If it is difficult to demonstrate experimentally, it would be good to present such content in the related work section.

Author Response

Please see the response letter to all comments in the attached file.

Reviewer 2 Report

-The motivation is insufficient. The challenges of data and device heterogeneity, as well as limited bandwidth resources, have already been extensively addressed in the existing literature, rendering them not novel. It is essential to clearly outline the approaches taken by previous researchers to tackle these challenges and identify the limitations or shortcomings of their solutions. This aspect needs further clarification to enhance the clarity of the manuscript.

-Why did you choose to investigate model pruning and parameter quantization as potential solutions for addressing the challenges of data and device heterogeneity, as well as limited bandwidth resources? From my understanding, there are multiple approaches available to tackle these challenges. Could you please provide some justification or rationale for focusing specifically on model pruning and parameter quantization in your study?

-Could you please provide more information about Proposition 1 and its relationship to the convergence of your proposed scheme? I would like to understand the theoretical basis or rationale behind Proposition 1 and how it guarantees convergence in the context of your approach. Clarifying this connection would enhance the comprehension of your manuscript.

-Based on Figure 3, it appears that the performance of your proposed scheme falls short compared to the baseline of FedAvg. Could you please provide an explanation for this disparity? Additionally, it would be helpful to highlight the advantages of your proposed scheme in comparison to other baselines. Furthermore, I would like to understand the costs associated with implementing your scheme. Please provide some insights into the trade-offs and considerations involved.

The quality of English language in this work is generally good. However, there are a few instances where the sentence structure could benefit from revision to enhance clarity and coherence. It is recommended to pay closer attention to subject-verb agreement, proper use of tenses, and overall sentence structure. These improvements will further enhance the clarity and readability of the manuscript.

Author Response

(The authors gave the same response as above.)

Round 2

Reviewer 2 Report

The authors have addressed my previous concerns.